# Lysyl Oxidase Family Enzymes and Their Role in Tumor Progression

**DOI:** 10.3390/ijms23116249

**Published:** 2022-06-02

**Authors:** Tanya Liburkin-Dan, Shira Toledano, Gera Neufeld

**Affiliations:** Cancer Research and Vascular Biology Center, The Bruce Rappaport Faculty of Medicine, Technion, Israel Institute of Technology, Haifa 31096, Israel; tili4light@gmail.com (T.L.-D.); shiratoledano1@gmail.com (S.T.)

**Keywords:** lysyl oxidases, tumor progression, extracellular matrix, epithelial to mesenchymal transition, angiogenesis

## Abstract

The five genes of the lysyl oxidase family encode enzymes that covalently cross-link components of the extracellular matrix, such as various types of collagen and elastin, and, thus, promote the stabilization of extracellular matrixes. Several of these genes, in particular lysyl oxidase (LOX) and lysyl oxidase like-2 (LOXL2) were identified as genes that are upregulated by hypoxia, and promote tumor cells invasion and metastasis. Here, we focus on the description of the diverse molecular mechanisms by which the various lysyl oxidases affect tumor progression. We also describe attempts that have been made, and are still on-going, that focus on the development of efficient lysyl oxidase inhibitors for the treatment of various forms of cancer, and of diseases associated with abnormal fibrosis.

## 1. Introduction

The lysyl oxidase (LOX) enzyme was identified as a secreted enzyme that catalyzes the deamination of the ε-amino group of lysines in collagen and elastin monomers, resulting in the formation of covalent cross-linkages, and the stabilization of collagen and elastin fibers. This is the first step in the cross-linking reaction of collagen and elastin [1,2]. Many years later, it was realized that LOX is but one member of a lysyl oxidase enzyme family that includes four additional members (LOXL1–4). The lysyl oxidases also produce hydrogen peroxide as a side product of their enzyme activity, and hydrogen peroxide, in turn, can activate signal transduction via the focal adhesion kinase (FAK) pathway [3,4]. The five lysyl oxidase family members are characterized by their highly conserved C-terminal regions, which contain their catalytic domains. The catalytic domains of all lysyl oxidases contain a copper-binding domain that is required for their catalytic activity, and a unique lysyl tyrosine quinone (LTQ) covalently linked internal structure that is also essential for their catalytic function [5]. However, the N-termini of the lysyl oxidases are much more diversified, and based upon their structure, the lysyl oxidases can be further divided into two subfamilies. The first subfamily includes LOX and LOXL1, which are synthesized as pro-enzymes that are cleaved by the bone morphogenic protein-1 (BMP-1) protease, to generate the active mature enzymes [6,7]. The second subdivision includes LOXL2–4, which is characterized by the presence of four scavenger receptor cysteine-rich (SRCR) domains in their N-terminal regions (Figure 1). It should be noted that there may be additional enzymatic functions associated with the SRCR domains of these lysyl oxidases. LOXL3 was found to associate with Stat3 in cell nuclei, and to deacetylate and deacetyliminate Stat3 on multiple acetyl–lysine sites. This activity was found to be associated with the N-terminal SRCR repeats of LOXL3, rather than with the C-terminal oxidase catalytic domain. Furthermore, when overexpressed, SRCR repeats from additional LOX family members also catalyze protein deacetylation/deacetylimination [8].

## 2. Enhancement of Tumor Progression by Lysyl Oxidases

Lysyl oxidases were first found to promote tumor progression in two independent manuscripts, published at about the same time. The first demonstrates that overexpression of LOX or LOXL2 enhances the invasion of breast cancer cells in in vitro invasion assays [9], and the second shows that overexpression of LOXL2 in breast cancer cells increases their invasiveness in a breast cancer mouse tumor progression model, and induces desmoplasia and the deposition of thick collagen bundles in tumors containing cells that strongly express LOXL2 (Figure 2) [10]. The pro-tumorigenic effects of these lysyl oxidases are not confined to breast cancer. LOX promotes the progression of colorectal cancer [11,12], cervical cancer [13], ovarian cancer [14], lung cancer [15,16], gastric cancer [17], and renal cell carcinoma [18], to cite some examples. Similarly, LOXL2 is found to promote colorectal cancer [19], gastric cancer [20,21], esophageal squamous cell carcinoma [22], cholangiocarcinoma [23], hepatocellular carcinoma [24], lung squamous cell carcinoma [25], non-small cell lung cancer [26,27], and renal cell carcinoma [28], to name some examples out of the many types of cancer in which tumor progression is enhanced by LOXL2. These studies were subsequently extended to show that additional lysyl oxidases also promote tumor progression. Thus, LOXL3 was observed to promote the progression of melanoma, and cooperates with BRAF in melanocyte transformation [29,30], and is also reported to promote the invasiveness and metastasis of breast cancer cells [31]. LOXL4 is reported to promote the proliferation and metastasis of gastric cancer [32], as well as hepatocellular carcinoma invasion and metastasis [33]. In another study, splice forms of LOXL4 were found to promote tumor progression in xenograft mouse models [34].

## 3. Extracellular Mechanisms by Which Lysyl Oxidases Promote Tumor Progression

### 3.1. Modification of Extracellular Matrix Components

The one structural element shared by the different lysyl oxidases is the catalytic domain located at their C-termini, which catalyze the oxidation of epsilon-amino moieties of lysines on extracellular matrix constituents, such as elastin and collagens. The expansion of solid tumors is frequently associated with the development of hypoxic regions within tumors, as the growth rate of new blood vessels cannot cope with the rapid rate of tumor expansion. This results in the upregulation of the expression of an array of hypoxia-induced genes, such as the proangiogenic factor VEGF [35,36]. Among these are also lysyl oxidases such as LOX and LOXL2, which are strongly upregulated by hypoxia [37]. One of the first things that was noticed when the link between LOXL2 overexpression and induction of tumor cells invasiveness was first established is that LOXL2 overexpression is associated with the deposition of a high concentration of collagen fibers in the tumor microenvironment. Furthermore, the deposited collagen fibers formed thick dense bundles, a hallmark of desmoplasia [10]. Desmoplasia was already known to be associated with tumor cells invasion in several types of tumors, most notably in breast cancer tumors [38,39]. These observations suggest that that the hypoxia-driven overexpression of lysyl oxidases by breast cancer tumor cells enhances tumor desmoplasia extracellular matrix stiffness, which, in turn, somehow enhances tumor cells invasiveness and tumor metastasis. It was then found that the enhanced cross-linking of collagen by lysyl oxidase induces integrin clustering in focal adhesions that, in turn, enhances PI3K signaling, which subsequently induces tumor cells invasiveness [40,41]. In addition, it was observed that enhanced expression of LOX in breast cancer cells activates the secreted protease HTRA1, which inhibits TGF-β1 signaling. This, in turn, increases the expression of matrilin-2, which subsequently inhibits the internalization of EGF receptors, causing an increase in their cell surface concentration. The increased cell surface concentration of these receptors results in enhanced sensitivity to EGF family ligands, which, in turn, enhances tumor growth and tumor metastasis [42]. Following these observations, additional evidence has accumulated suggesting that lysyl oxidases, such as LOXL2 and LOX, promote desmoplasia under various conditions and, as a result, promote tumor cells invasiveness and tumor metastasis [43,44,45,46,47]. Furthermore, interference with collagen stabilization by the inhibition of lysyl oxidases significantly enhances responses to chemotherapy in various tumor models, even in metastatic disease [48].

Secreted lysyl oxidases, such as LOX and LOXL2, also influence the spread of tumor cells to distant sites by another mechanism. Lysyl oxidases secreted from cells within primary tumors enter the circulation as soluble enzymes, or packaged in extracellular secreted vesicles. When they reach distant organs, they can then modify the extracellular matrix at these distant sites, rendering these distant locations more permissible for colonization by metastasizing tumor cells [49,50,51].

### 3.2. Modulation of the Focal Adhesion Kinase (FAK) Signaling Pathway by Lysyl-Oxidases

A by-product of the classic lysyl oxidase enzyme activity is hydrogen peroxide. Hydrogen peroxide produced as a result of LOX activity activates signal transduction via the FAK/src pathway, and the activation is abrogated when hydrogen peroxide is removed by catalase. Increased src phosphorylation is known to be correlated with enhanced metastasis [3,15,52,53]. Likewise, LOXL4 expression is found to be upregulated in hepatocellular carcinoma, and is associated with enhanced tumor cells invasiveness and tumor metastasis to lungs. Interestingly, it was observed that hepatocellular carcinoma-derived exosomes transferred LOXL4 between hepatocellular carcinoma cells. It was observed that intracellular LOXL4, rather than extracellular LOXL4, promoted cell migration through the activation of the FAK/Src pathway, through the upregulation of hydrogen peroxide production [33].

### 3.3. Pro-Angiogenic Effects of Lysyl Oxidases

The enzyme activity of the lysyl oxidases is copper-dependent. It was observed that reduction in copper availability can inhibit tumor angiogenesis [54], suggesting that inhibition of lysyl oxidases enzyme activity through copper deprivation may perhaps be partly responsible for this observation. Indeed, LOX functions as a promoter of angiogenesis [55,56], and was observed to enhance tumor angiogenesis in several types of cancer [56,57]. LOX expression is found to be correlated with increased vascular endothelial growth factor (VEGF) and platelet-derived growth factor expression. Indeed, conditioned media obtained from LOX overexpressing hepatocellular carcinoma cells stimulates angiogenesis via enhanced VEGF production, and enhances the tube formation capacity of endothelial cells. Similarly, it is observed that the interplay between VEGF, TGFß, and LOX is essential for the formation of mature vascular, smooth muscle cells-coated vessels [58]. LOX expression increases extracellular matrix stiffness and this, in turn, also enhances the expression of VEGF, as well as platelet-derived growth factor (PDGF) by tumor resident cells [59,60]. Interestingly, it is reported that LOX expression is also enhanced in endothelial cells of tumor-associated blood vessels, and that the produced LOX further enhances tumor angiogenesis [61]. It should be noted however, that LOX-PP has an opposite effect on angiogenesis [62]. LOXL2 also promotes angiogenesis, since it is required for the correct assembly of collagen-IV, which is the major constituent of the basement membrane that underlies sheets of endothelial cells in blood vessels [63]. Similar effects were also associated with LOXL4 expression [64], and likewise, it is reported that LOXL1 promotes angiogenesis [65]. Interestingly, the pro-angiogenic activity of LOXL2 also does not seem to require the lysyl oxidase enzyme activity, and seems to be mediated by the LOXL2 SRCR domains [66]. It may also induce angiogenesis by enhancing epithelial to mesenchymal transition of endothelial cells [67], as well as by the modulation of gene expression via LOXL2-induced changes in histone methylation [68]. It is demonstrated that inhibition of LOXL2 inhibits tumor angiogenesis in several types of solid tumors [69,70,71]. Interestingly, it is also reported that LOXL2 induces vasculogenic mimicry [72] in hepatocellular carcinoma [73].

## 4. Intracellular Mechanisms by Which Lysyl Oxidases Enhance Tumor Progression

Epithelial cells are usually confined to epithelia and are relatively non-motile. Tumors that arise from such cells are initially confined to the tissue of origin until some of the transformed cells undergo epithelial to mesenchymal transition that enables them to invade neighboring tissues [74,75,76]. It is initially observed that the overexpression of either LOXL2 or LOXL3 induces epithelial to mesenchymal transition. It was first reported that these lysyl oxidases oxidize lysine residues 98 and 137 of the transcription factor snail, which, as a result, leads to ubiquitin-mediated degradation of snail. This, in turn, inhibits the expression of E-cadherin, which is a major trigger of epithelial to mesenchymal transition [31,77]. However, it is noted in a subsequent publication that LOXL2 point mutants lacking catalytic activity also induce EMT and that it is the physical association of LOXL2 with snail that triggers LOXL2-induced epithelial to mesenchymal transition [78,79]. Indeed, a natural catalytically inactive LOXL2 splice forms, which lacks the classical lysyl oxidase enzyme activity, fails to be secreted, and accumulates in the cytoplasm, and enhances the invasive activity of esophageal squamous carcinoma cells [80].

It was also observed that LOXL2 interacts with myristoylated, alanine-rich C kinase substrate-like 1 (MARCKSL1), via its SRCR domains. This, in turn, inhibits MARCKSL1-induced apoptosis. It is also reported that LOXL2 promotes oncogenic tumor progression in alveolar rhabdomyosarcoma, independently of its catalytic activity, and possibly as a result of its interaction with the intermediate filament protein vimentin [81]. Lastly, both LOXL2 and its catalytically inactive splice form LOXL2delta13 interact with several actin-binding proteins, such as ezrin, fascin, and tropomodulin-3. Phosphorylation of ezrin is enhanced by both LOXL2 and the catalytically inactive LOXL2 splice variant. This, in turn, results in cytoskeletal reorganization that enhances tumor cell invasion and metastasis [82]. These reports as well as additional reports [83], suggest that LOXL2 activates multiple signaling pathways, independently of its classical amine oxidase activity, and that their activation promotes tumor progression. However, the LOXL2 classical amine oxidase activity is, nevertheless, required for some of its intracellular functions. Thus, it was found that LOXL2 is able to deaminate trimethylated H3K4, thus, indirectly modulating the expression of the cadherin-1 (CDH1) gene. CDH1 is a transcription factor for E-cadherin, which, when inhibited, promotes epithelial to mesenchymal transition [84].

Interestingly, in the case of LOXL3, it seems that in addition to its activity as an amine oxidase, it also functions as a deacetylase. LOXL3 associates with Stat3 in the nucleus, resulting in the deacetylation of Stat3 on multiple acetyl–lysine sites. Surprisingly, LOXL3 N-terminal scavenger receptor cysteine-rich (SRCR) repeats, rather than the C-terminal oxidase catalytic domain, function as the deacetylase enzyme activity containing domains. As a result, Stat3 dimerization is inhibited and its activity as a transcription factor abolished [8].

It was also observed that LOX re-enters producing cells and moves to their nuclei by mechanisms that are not yet completely clear [85]. It is suggested that LOX associates with the p66β transcription factor that apparently transports it to cell nuclei [86]. Similarly to LOXL2, LOX also induces epithelial to mesenchymal transition by intracellular activation of gene expression. Twist is a transcription factor that, when expressed, inhibits E-cadherin expression and, thus, promotes epithelial to mesenchymal transition [87]. LOX was found to be an essential component of the CD44–Twist signaling axis. Lox stimulates Twist expression under hypoxia, and thereby promotes tumor metastasis [17,88,89]. LOX can also be translocated to cell nuclei [90,91,92] where it binds to the promoter of the SNAI2 (SLUG) gene, which is also a down-regulator of E-cadherin [93], and stimulates SNAI2 expression. This, in turn, promotes the expression of tissue inhibitor of metalloproteinases-4 (TIMP-4), which also functions as an enhancer of epithelial to mesenchymal transition [94,95]. Interestingly, histone-H1 was also identified as a substrate for LOX [96]. This was observed to promote histone-H1 detachment from DNA, and consequently opens the mouse mammary tumor virus (MMTV) promoter structure to activating transcription factors [97].

There are also some reports that characterize LOX as a repressor of oncogenic transformation. Thus, the oncogenic transformation of Ewing sarcoma is induced by the expression of the ETS-type fusion oncoprotein EWS/FLI, which functions as a transcriptional activator. It turns out that LOX is a EWS/FLI-repressed target that inhibits the transformed phenotype of Ewing sarcoma cells [98]. It is not clear if, in this case, the repression is due to the activity of LOX-PP.

## 5. Lysyl Oxidases and Immune Responses

Not much has been published regarding the role of lysyl oxidases produced by immune cells on tumor progression, but the little that is published suggests that lysyl oxidases produced by such cells may contribute to tumor progression in various ways. The expression of LOXL3 is found to be upregulated in hepatocellular carcinoma tumors. The enhanced expression of LOXL3 is correlated with the enhanced infiltration of multiple immune cells, and with the expression of immune checkpoint genes in these tumors [99]. Injection of LOXL4 promotes the infiltration of macrophages into the liver, and accelerates tumor growth [100]. Similarly, LOX inhibition markedly suppresses macrophage infiltration and glioma tumor progression. In PTEN-deficient glioma tumors, YAP1 is activated, and YAP1-LOX and beta1 integrin-SPP1 signaling correlate positively with higher macrophage density in glioblastoma tumors, and with lower overall survival in glioblastoma multiforme patients [101]. Pancreatic ductal adenocarcinoma tumors are characterized by an expansion of B-lymphocyte subsets that express LOXL2, and thereby contribute to the stromal reaction in these tumors [102]. LOX expression is also significantly induced in human leukemic THP-1 cell-derived M2-like macrophages. This is accompanied by a decrease in the level of histone H3 tri-methylation at lysine 27. Pre-treatment with a H3K27 demethylase inhibitor suppresses the expression of LOX in these cells. LOX derived from M2-like macrophages is also found to enhance breast cancer cells migration, and the LOX-induced migration is, likewise, inhibited by a H3K27 demethylase inhibitor [103].

Lysyl oxidases produced by primary tumors are found to promote extracellular matrix remodeling in distant sites such as lungs and, as a result, make these sites more receptive to colonization by metastatic tumor cells [49,50]. However, recent studies indicate that immune cells that express lysyl oxidases may also contribute to the remodeling of pre-metastatic niches. Thus, treatment of healthy mice with the chemotherapeutic agent paclitaxel induced extracellular matrix remodeling in their lungs. The remodeling was the result of chemotherapy-induced infiltration of CD8 + T cells expressing LOX. Treatment of mice harboring metastatic breast carcinoma tumors with paclitaxel enhances the metastasis of the tumor cells to lungs. The metastatic process is inhibited by a LOX inhibitor, suggesting that chemotherapeutic treatment enhances metastasis through the recruitment of LOX-expressing immune cells, which turn pre-metastatic niches, such as lungs, more receptive to colonization by metastatic tumor cells [47].

## 6. Effects on Tumor Dormancy

Healthy individuals frequently harbor dormant tumors that remain dormant for many years [104]. Frequently, tumors contain sub-populations of semi-quiescent cancer cells that evade treatment with chemotherapeutic agents, leading to tumor relapse [105]. Finally, primary tumors frequently suppress the development of distant metastases, which develop following the excision of the primary tumor [106]. The mechanisms that govern the transition from dormancy to active growth are diverse, and it seems that lysyl oxidases play a role in this process too. Thus, dormant MCF-7 breast cancer cells colonizing the lung remain dormant, but transition to metastatic growth following LOXL2 expression. These findings are further supported by clinical data demonstrating that increases in LOXL2 mRNA levels correlate with increases in epithelial to mesenchymal transition and stem cells markers, and LOXL2 expression is also associated with a decrease in relapse-free survival of breast cancer patients [107].

## 7. Inhibition of Tumor Progression by Lysyl Oxidases

Although much less frequent, there are also some reports in which lysyl oxidases are associated with the inhibition of tumor progression. Perhaps the best characterized example is LOX. Following cleavage by BMP-1, the cleaved N-terminal part, also known as pre-pro LOX (LOX-PP), functions as an inhibitor of tumor progression [108,109]. In early publications, lysyl oxidase was described as a *ras* recision gene because it inhibits ras-induced transformation of cells [110,111]. However, there are also a few reports that indicate that additional lysyl oxidases may sometimes inhibit tumor progression. Thus, downregulation of LOXL2 expression is found to be associated with disease progression in lung adenocarcinomas [112], and the inhibition of LOXL4 expression by miR-135a or miR-210 is found to promote, rather than repress, the progression of lung cancer [113,114]. Similar results are reported for LOXL4 in a xenograft breast cancer model [115]. Additionally, it was found that LOXL4 expression is upregulated by 5-azacytidine. Upregulated LOXL4 is found to associate with p53 in nuclei, thereby inducing the reactivation of compromised p53, resulting in cell death. Experiments in nude mouse xenograft models show that the LOXL4/p53 interaction reduces tumor growth, and in liver cancer patients, LOXL4 expression is correlated with overall survival [116]. The LOXL4 mRNA is found to be alternatively spliced, and two splice forms were identified. Interestingly, the product of the full length mRNA is found to behave as a tumor suppressor, while the products of the alternatively spliced forms function as promoters of tumor progression [34], which may explain the contradictory reports concerning the effects of LOXL4 on tumor progression. Finally, LOXL1 is found to function as a suppressor of colorectal cancer [117].

## 8. Development of Anti-Tumorigenic Therapeutics Targeting Lysyl Oxidases

As detailed above, many studies show that lysyl oxidases, in particular LOX and LOXL2, promote the progression of many types of tumors. This implies that inhibitors of lysyl oxidases may turn out to be useful therapeutic agents for the treatment of many types of cancer. In addition, inhibition of lysyl oxidases may assist in the control of fibrotic diseases, such as idiopathic lung fibrosis and liver cirrhosis, and may help to reduce tumor desmoplasia [118,119,120]. The first attempt in this direction was the development of a LOXL2-targeting antibody that showed promise in pre-clinical studies [43,121,122]. However, a humanized version of this antibody, named simtuzumab, failed to show clinical effectiveness in several clinical trials that targeted several types of cancer and fibrotic diseases [123,124,125]. The causes of this failure are unclear. One plausible explanation is that the antibody was too specific, and that lysyl oxidases other than LOXL2 compensated for LOXL2 inhibition. Lastly, since the antibodies are not internalized effectively, it probably failed to inhibit LOXL2 intracellular activities, such as the induction of epithelial to mesenchymal transition, which seems to be independent of the classical lysyl oxidase enzyme activity [78,80]. Another antibody that targets the active site of LOXL2, GS341, inhibits the development of tumors from MDA-MB-231 breast cancer cells, and inhibits their metastatic spread to lungs, but has not yet evaluated in clinical trials [126,127].

In view of these observations, attempts are currently being made that aim to develop agents that will address at least some of these obstacles. β-aminopropionitrile (BAPN) is a small, irreversible inhibitor of the enzyme activity of all lysyl oxidases with particular affinity to LOXL2 [128,129]. However, it fails to inhibit tumor progression in a mouse model of prostate cancer [130]. A different class of small-molecular-weight inhibitors based upon the structure of haloallylamine was recently described [65]. These early inhibitors were recently further refined, and the second generation small-molecular-weight LOXL2 inhibitors, PXS-5338, PXS-5382, and PXS-5878, were recently reported to inhibit the enzyme activity of LOXL2 more effectively that simtuzumab [131]. In addition, the same company developed a pan-LOX inhibitor, PXS-5505, which is supposedly able to inhibit the activities of several lysyl oxidases [132], and was reported to be safe in a phase 1 clinical study [133]. It remains to be seen if these inhibitors will prove to be more effective in further clinical trials.

Another small-molecular-weight agent, PAT-1251, was also developed as a LOXL2 inhibitor, ref. [134], but much less is published regarding its properties. Recently, epigallocatechin gallate (EGCG) was characterized as a LOXL2 and TGF-β1 inhibitor, and is in early clinical trial [135].

## 9. Conclusions

Overwhelming evidence has accumulated suggesting that the expression of several lysyl oxidases, in particular LOX and LOXL2, is enhanced in the hypoxic tumor microenvironment. The enhanced expression is correlated with enhanced invasiveness, and promotes tumor metastasis. Enhanced expression of lysyl oxidases is also found to play a central role in other diseases characterized by abnormal fibrosis. It was thought that lysyl oxidases induce tumor cells invasiveness and metastasis due to their extracellular collagen cross-linking activity, which leads to extracellular matrix stiffening. However, there is more and more evidence indicating that lysyl oxidases also promote tumor cells metastasis by targeting intracellular proteins, such as histones or transcription factors, such as snail, in the cases of LOXL2 and LOXL3. Furthermore, the classical oxidase activity is not required for the modulation of the activity of some of these intracellular signaling pathways by lysyl oxidases. Taken together, these observations present problems for developers of therapeutic agents designed to inhibit tumor progression by the targeting of lysyl oxidases. If we take inhibition of LOXL2 as an example of a potential target, the inhibitor needs to be able to be internalized, and block both the oxidase-dependent and the oxidase-independent pro-tumorigenic activities of LOXL2. This could perhaps be circumvented by the simultaneous use of several therapeutic agents targeting different domains of interest. However, this still leaves open the possibility of compensatory effects by other lysyl oxidases, which can be produced by the cancer cells, or by other cell types attracted to the tumor microenvironment. Thus, efficient inhibition of tumor progression may require the simultaneous inhibition of the activities of more than a single lysyl oxidase.

## Figures and Tables

**Figure 1 ijms-23-06249-f001:**
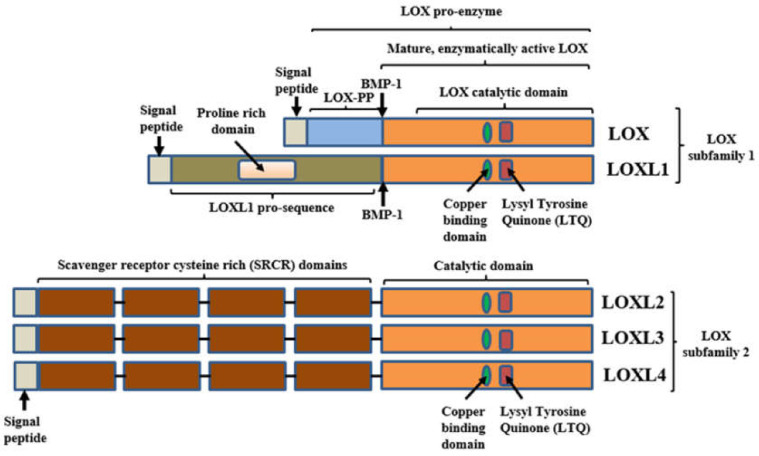
Structural elements of the lysyl-oxidases: The five lysyl-oxidases can be divided into two sub-families. The first includes LOX and Loxl1. These are synthesize as pro-enzymes that are cleaved by the BMP-1 protease (and in the case of LOX by additional proteases such as ADAMTS2/14) to release the mature active enzyme. The second subfamily includes LOXL2-4 and is characterized by a much less conserved N-terminal that is characterized by the presence of four scavenger receptor cysteine rich (SRCR) domains. The catalytic domain is highly conserved among the different lysyl-oxidases and includes a conserved copper binding domain and a unique lysyl tyrosine quinone (LTQ) element that is essential for the catalytic activity (for a more detailed discussion of the structure and enzyme function of LOX see this review [5]). In the case of LOX cleavage by BMP-1 also releases the N-terminal part, LOX-PP, which functions as a tumor suppressor.

**Figure 2 ijms-23-06249-f002:**
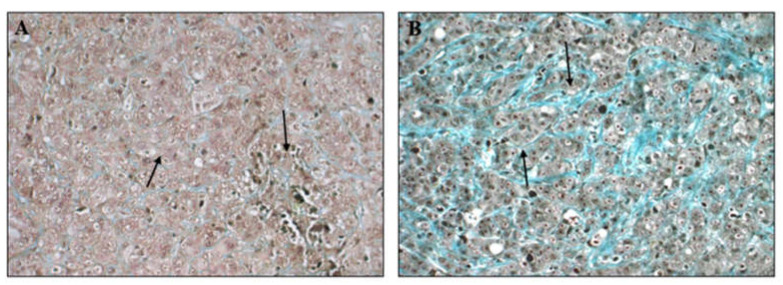
LOXL2 expression promotes the deposition of collagen fibers in tumors: MCF-7 breast cancer derived cells were transfected with an empty expression vector or a vector directing expression of the full length cDNA encoding LOXL2. (**A**) The living parts of tumors derived from MCF-7 cells transfected with empty expression vector which do not suffer from excessive hypoxia contain low concentrations of collagen fibers as revealed by blue Mason’s trichrome staining (arrows). (**B**) In contrast, the living parts of tumors derived from MCF-7 cells expressing recombinant LOXL2 contain much higher concentrations of collagen fibers (arrows) (Pictures taken from reference [10]).

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
