# Peer review of "Lysyl Oxidase Family Enzymes and Their Role in Tumor Progression"

_ijms, 2022, doi:10.3390/ijms23116249_

Round 1
Reviewer 1 Report
Lysyl Oxidase family enzymes and their role in tumor progression
by Tanya Liburkin-Dan, Shira Toledano and Gera Neufeld
Liburkin-Dan et al., aim to provide a detailed review dealing with the functional role of lysyl-oxidase enzymes during tumor progression.
However, the reviewer has major concerns with the presented manuscript and can therefore not recommend publication in ist current form.
The biggest concern ist hat the current review is not needed in ist current form because there are no further insights compared to other reviews collecting the knowledge about lysyl-oxidase family members.
Furthermore, this review is not properly structured and reveals only a list of findings in ist current form. I strongly recommend to write a coherent review with a clear structure and giving some vision from the authors why discrepancies can be found in the literature.
Please prevent to only offering the reader a list of observations one by one.
There is an urgent need for a thorough improvement of the language. There are several grammar and spelling mistakes.
Minor points and just stop to improve the language.
Introduction:
Line 10: „…was identified as a secreted…“
Line 11: „…of lysines in collagen and…“
Line 14: „…it has been realized that LOX is one member…“ (delete but)
Line 22: „However, the N-terminus…“
Line 79: either „…their C-termini which catalyse…“ or „…their C-terminus which catalyses…“
Reviewer 2 Report
Summary:
As indicated by the title, the authors have compiled a review of the literature regarding the role of the Lysyl Oxidase (LOX) family and their role in cancer. The authors have a track record of publishing in the field of cancer and LOX family protein function, and are therefore well placed to write a such a review. This topic is an active area of research and this review places new knowledge within an historical context. I enjoyed reading the article and especially the inclusion of information about the intracellular functions of this family of proteins. I am therefore supportive of publication in the current format but have some suggestions that the authors may wish to consider.
Comments:
- This reviewer was previously ignorant of the intracellular functions of LOX family proteins. I therefore noticed the statement at line 16-17 regarding the role of LOX activity in H2O2 production and FAK activity and wanted to know if this was outside or inside the cell. Upon further reading of your manuscript, the additional intracellular roles for LOX family proteins were discussed, but I feel a clear statement of both the extracellular and intracellular roles of these proteins would help the reader early in the introduction and abstract sections.
- Figure 1 indicates the presence of signal peptides in all LOX family members but how does this fit with their intracellular and extracellular localisation?
- Section 2 Line 57: How does the reported roles of LOX in tumour progression with role of desmoplasia in many cancers? It is known that collagen production from cancer fibroblasts have both tumour promoting and inhibitory roles. Are LOXs the same? Does LOX expression and function correlate with desmoplasia? Also what about LOX in terms of immune hot or cold tumours too? Does LOX expression function stop or promote certain immune cell retention in tumours?
- In general, for many of the examples provided it may be worth stating more clearly which cell types secrete LOX in the context of the tumour microenvironment?
- In terms of the review structure, is section 3 better placed as part of section 2? Perhaps you could consider a review of your section headings? The heading for section 3.5 in particular would benefit from rephrasing.
- There a several small grammatical / typographical errors e.g. line 10 should read ‘as a secreted enzyme’, line 75 heading needs capitalisation of ‘Extracellular’, line 108 needs capitalisation of ‘Secreted’. Please check all sections for similar minor issues.
- Line 151. Please use the correct nomenclature for collagen IV
- The abstract was missing from the pdf for review.
- Figure 2 was not referred to in the manuscript text and it was not clear how and where it would fit.
- Whilst checking for similar recent reviews on this topic in the literature I found the following recent manuscript that may be worth including: Amendola, P.G.; Reuten, R.; Erler, J.T. Interplay Between LOX Enzymes and Integrins in the Tumor Microenvironment. Cancers 2019, 11, 729. https://doi.org/10.3390/cancers11050729
Round 2
Reviewer 1 Report
Thanks to the authors to add more knowledge from the recent literature.
Please consider in implementing a vision and some ideas about
-compensatory effects
-how can we improve targeting
Finally, the review still looks like a list quite comprehensive and still lacking a nice line.
Author Response
The reviewer states that extensive editing of English language and style is required. We did that after the first review. With all due respect, I have written many reviews during my thirty year long scientific career. Some of these such this one (Neufeld, G. et al (1999) Vascular endothelial growth factor (VEGF) and its receptors. FASEB J. 13, 9-22) are very highly cited. I have never had any complaints about English style or grammar and since they are highly cited I assume that the readers did understand their content and were not bothered by any language problems. The reviewer did not point out any specific problems in the revised review. If he will point them out we will be happy to correct. The second reviewer did not raise any concerns with regard to the revised version but if you are not satisfied yet than I suggest that you send the review to a third reviewer and see what he suggests.
The reviewer also criticizes the organization of the revised review but again, does not specify what bothers him about the organization. Thus I do not know how to answer this critique.
Thanks to the authors to add more knowledge from the recent literature.
Reply: We understand from this that the reviewer is satisfied with the changes that we made.
Please consider in implementing a vision and some ideas about
-compensatory effects
Reply: I assume that by this remark the reviewer relates to the compensatory effects caused by the overlap in the functions of the different lysyl-oxidases. The lysyl-oxidases share very similar biological activities, and as a result, targeting only one lysyl-oxidase family member may not result in an effective anti-tumorigenic outcome since other lysyl-oxidases may compensate. We have highlighted this problem in our conclusions and have also mentioned that a drug that aims to inhibit the enzyme activities of all the lysyl-oxidases is under development at this time. Nevertheless, we have now further modified the conclusions section in order to highlight the problems that developers of therapeutics targeting lysyl-oxidases may have to overcome.
-how can we improve targeting
The answer to that is that I simply do not know. Maybe there will be a reader that will have some novel ideas.
Finally, the review still looks like a list quite comprehensive and still lacking a nice line.
Reply: I do not understand what the reviewer means by "lacking a nice line". In this review we wanted in particular to draw attention to the manuscripts that suggest that several lysyl-oxidases can also promote or inhibit tumor progression by mechanisms that are independent of their classical oxidase activity. In addition we also wanted to highlight lysyl-oxidase mediated effects on tumor progression that take place in intracellular compartments and do not seem to require the secretion of these lysyl-oxidases. This data is frequently conveniently ignored when inhibition of tumor progression through inhibition of lysyl-oxidases is discussed. However, this is a major point of concern for the development of therapeutic agents that target lysyl-oxidases since it means first that inhibitors of enzyme activity may not suffice, and furthermore, that the proposed drugs may need to be internalized into the cancer cells to be effective. We have re-written the conclusions section in order to focus on these issues and we hope that it is now clearer. We hope that these changes answer to concerns of the reviewer.
Round 3
Reviewer 1 Report
Line 24: "was identified as a..."
Line 53: "...in in vitro..."
Line66: also is doubled! Please adjust
Paragraph 2 (Enhancement of tumor progression by lysyl-oxidases): This paragraph is clearly a list of examples. What do you think? Do here all LOX proteins do similar things? What is your view? Any idea? Can here LOX proteins compensate each other?
Line 76: Here, you use termini and in line 36 you use terminuses. Please be consistent.
Line 76: "...which catalyzes..."
Line 83: "...One of the first things that were noticed when the link between LOXL2 83 over-expression and induction of tumor cells invasiveness was first noticed was that..." Is this sentence correct?
Line 94: "...that in turn enhances..., which in turn..." Try to reformat the sentence and not always using "in turn"
Line 113: A byproduct
Line 120: "...carcinoma-derived..."
Line 127: "...may perhaps..." This can be framed with just using may as may includes perhaps by definition
Line 135: ",and LOX..." to ", and LOX..."
Line 141: Define LOX-PP
Line 146: what to you want to express with the word "too"?
Line 171: "...Promotes..." change to "...promotes..."
Line 172: why is catalytic activity written in capital letters?
Line 176 and 177: Again two times in turn. Please reformat.
Please reformat the following part as independent of classical is mentioned twice.
These reports and additional reports that have identified LOXL2 biological 178 functions independent of the classical lysyl-oxidase activity [83], suggest that LOXL2 179 may activate multiple signaling pathways independently of its classical amine-oxidase 180 activity, and that their activation can promote tumor progression.
I am sorry but I strongly believe that proper editing is not my duty. Therefore, please go over again.
I am still puzzled, if the responsibility of proper editing is the duty of the reviewer or the authors. Do you think that a reviewer has to correct all English grammar mistakes?
Line 298: "...which may not have been insufficient to produce..." Is this correct?
Line 313 to 316: Is this sentence correct?
Line 319: Why is the word but small?
I am aware that the authors have written highly cited reviews and I am sorry that I have to express my concerns again.
I hope that some of my suggestions (not mentioned all) will help to improve the quality of this review.
Authors response to Comments and Suggestions for Authors from reviewer 1 in last round of review
Line 24: "was identified as a..."
Reply: We corrected that
Line 53: "...in in vitro..."
Reply: I do not understand this critique. Seems correct to me
Line66: also is doubled! Please adjust
Reply: I do not see anything wrong with line 66. I do not know what the reviewer wants.
Paragraph 2 (Enhancement of tumor progression by lysyl-oxidases): This paragraph is clearly a list of examples. What do you think? Do here all LOX proteins do similar things? What is your view? Any idea? Can here LOX proteins compensate each other?
Reply: We have related to all of these issues in the paragraphs subsequent to paragraph 2. Indeed in this paragraph we only wanted to present at least a partial list of all the cancer subtypes in which lysyl-oxidases were found to play a role. We think this is the proper way to organize the review.
Line 76: Here, you use termini and in line 36 you use terminuses. Please be consistent.
Reply: Corrected to termini
Line 76: "...which catalyzes..."
Reply: corrected
Line 83: "...One of the first things that were noticed when the link between LOXL2 83 over-expression and induction of tumor cells invasiveness was first noticed was that..." Is this sentence correct?
Reply: The sentence is now correct
Line 94: "...that in turn enhances..., which in turn..." Try to reformat the sentence and not always using "in turn"
Reply: The sentence was corrected
Line 113: A byproduct
Reply: Corrected
Line 120: "...carcinoma-derived..."
Reply: Corrected
Line 127: "...may perhaps..." This can be framed with just using may as may includes perhaps by definition
Reply: We have now modified this sentence
Line 135: ",and LOX..." to ", and LOX..."
Reply: Corrected
Line 141: Define LOX-PP
Reply: LOX-PP is defined in the legend of figure 1
Line 146: what to you want to express with the word "too"?
Reply: We re-wrote this sentence to clarify.
Line 171: "...Promotes..." change to "...promotes..."
Reply: Corrected
Line 172: why is catalytic activity written in capital letters?
Reply: Corrected
Line 176 and 177: Again two times in turn. Please reformat.
Reply: Corrected
Please reformat the following part as independent of classical is mentioned twice. 2 These reports and additional reports that have identified LOXL2 biological 178 functions independent of the classical lysyl-oxidase activity [83], suggest that LOXL2 179 may activate multiple signaling pathways independently of its classical amine-oxidase 180 activity, and that their activation can promote tumor progression.
Reply: We re-wrote this sentence.
I am sorry but I strongly believe that proper editing is not my duty. Therefore, please go over again.
I am still puzzled, if the responsibility of proper editing is the duty of the reviewer or the authors. Do you think that a reviewer has to correct all English grammar mistakes?
Line 298: "...which may not have been insufficient to produce..." Is this correct?
Reply: I thank the reviewer for noticing this mistake. The sentence was deleted.
Line 313 to 316: Is this sentence correct?
Reply: This is correct. This is what they claim. We cited the publication
Line 319: Why is the word but small?
Reply: I did not understand this comment.
I am aware that the authors have written highly cited reviews and I am sorry that I have to express my concerns again.
I hope that some of my suggestions (not mentioned all) will help to improve the quality of this review
I thank the reviewer for his careful editing. Some of his comments were very helpful. I agree that it is not the job of the reviewers to correct grammar mistakes and typos, but it's very helpful if they do.